# *Rhus coriaria* L. (Sumac), a Versatile and Resourceful Food Spice with Cornucopia of Polyphenols

**DOI:** 10.3390/molecules27165179

**Published:** 2022-08-14

**Authors:** Gaber El-Saber Batiha, Oludare M. Ogunyemi, Hazem M. Shaheen, Funso R. Kutu, Charles O. Olaiya, Jean-Marc Sabatier, Michel De Waard

**Affiliations:** 1Department of Pharmacology and Therapeutics, Faculty of Veterinary Medicine, Damanhour University, Damanhour 22511, AlBeheira, Egypt; 2Human Nutraceuticals and Bioinformatics Research Unit, Department of Biochemistry, Salem University, Lokoja 260101, Nigeria; 3Nutritional and Industrial Biochemistry Research Unit, Department of Biochemistry, University of Ibadan, Ibadan 200005, Nigeria; 4School of Agricultural Sciences, University of Mpumalanga, Mbombela 1200, South Africa; 5Institut de Neurophysiopathologie (INP), CNRS UMR 7051, Faculté des Sciences Médicales et Paramédicales, Aix-Marseille Université, 27 Bd Jean Moulin, 13005 Marseille, France; 6Smartox Biotechnology, 6 Rue des Platanes, 38120 Saint-Egrève, France; 7L’institut du Thorax, INSERM, CNRS, UNIV NANTES, 44007 Nantes, France; 8LabEx Ion Channels, Science and Therapeutics, Université de Nice Sophia-Antipolis, 06560 Valbonne, France

**Keywords:** sumac, spice, nutraceuticals, polyphenols, antioxidants, diabetes, additives

## Abstract

In recent years, utilization of *Rhus coriaria* L. (sumac) is upgrading not only in their culinary use and human nutrition, but also in the pharmaceutical industry, food industry and veterinary practices. This is driven by accumulating evidence that support the ethnobotanical use of this plant; in particular, advanced knowledge of the content of nutritional, medicinal and techno-functional bioactive ingredients. Herein, we discuss polyphenolic compounds as the main bioactive ingredients in *Rhus coriaria* L., which contribute mainly to the significance and utility of this spice. Most of the antioxidant potential and therapeutic roles of sumac are increasingly attributed to its constituent tannins, flavonoids, and phenolic acids. Hydroxyphenyl pyranoanthocyanins and other anthocynins are responsible for the highly desired red pigments accounting for the strong pigmentation capacity and colorant ability of sumac. Certain polyphenols and the essential oil components are responsible for the peculiar flavor and antimicrobial activity of sumac. Tannin-rich sumac extracts and isolates are known to enhance the food quality and the oxidative stability of animal products such as meat and milk. In conclusion, polyphenol-rich sumac extracts and its bioactive ingredients could be exploited towards developing novel food products which do not only address the current consumers’ interests regarding organoleptic and nutritional value of food, but also meet the growing need for ‘clean label’ as well as value addition with respect to antioxidant capacity, disease prevention, and health promotion in humans.

## 1. Introduction

Food spices and herbs are not only desired for their aromatic and culinary properties but also for their potentials in disease prevention, drug therapy, industry and agriculture. These dietary components have played key roles in the lifestyle of people of different cultures through history, as they are valued and often utilized as coloring agents, flavoring agents, preservatives, and therapeutic agents in food preparation. Spices could be produced from several plant parts including the roots, rhizomes, stem bark, leaves, fruits, flower and seeds [1]. Although, most spices are dried and used in a processed but complete state, aromatic leaves, which are commonly referred to as herbs are often utilized in fresh state. Spice products can also be produced through solvents extraction of the oleoresins and other standardized products. Extracts such as essential oils which are known to contain the aromatic and pungent principles are often produced by distilling the raw wet or dried sample of the plant material. Spices vary greatly in composition but the lachrymatory factors and the medicinal value that render them valuable reside in their bioactive constituents viz: volatile oils, resins, and phytochemicals.

Sumac (*Rhus coriaria* L), is a culinary spice that has been used for several purposes in the Mediterranean region and the northern parts of Africa since antiquity. This plant, which belongs to the Anacardiaceae family, is the most economically important species among over 250 members of the genus Rhus. The plant thrives well in subtropical and temperate climates, particularly in various parts of the Mediterranean, Asia, and northern Africa [2]. The plant is a 3–5 m tall shrub, with long leaves, having pinnate veins with 7–8 pairs of small oval leaflets of varying sizes (Figure 1). *Rhus coriaria* produces dark red drupes with thin flesh, which form dense clusters at branch tips, known as sumac bobs. The plant owes its name to the red color of the important and widely used spice product of the plant. It is believed by some people that, the word sumac originated from the Arabic word “summaq”, which translates to “dark red”, while some other people believe that it is derived from “Sumaga” a Syriac word meaning to “red” [3]. These berries are edible, tangy and delicious, containing malic acid which is found in apples [3,4]. Sumac has a long history of culinary and traditional uses in different cultures [5]. Although, the fresh fruits of sumac can be used to make tea, more often they are dried, crushed and blended to a thin red-purple powder for use as culinary seasoning. The powder is widely used as a condiment to be sprinkled on dishes and used in combination with salt and onions as a seasoning for roast meat throughout the Mediterranean region [6]. In northern Africa, sumac is often rubbed on meats, chicken or fish. It is commonly added to marinades and also used to improve the flavor and acidity of yogurt sauces or vinaigrettes. It can be added to egg dishes and salads in order to enhance their taste and flavor. Due to its attractive red color, sumac products are used as decorative garnishes on several dishes [4,7]. Recognition of the medicinal value of sumac can be traced back to about 2000 years ago when the Greek physician Pedanius Dioscorides (40–90 AD) mentioned it in his writings “*De Materia Medica*” talking about “Medical Issues” and the benefits of sumac, especially as an anti-flatulent and diuretic [8]. The bark powder of *Rhus coriaria* has been used as an effective teeth cleaning agent, while its infusion has been used to treat viral eye infections. The powdered spice product from the dried fruit was sparsed on boiled eggs and consumed for diarrhea treatment [9]. A fruits decoction is set and traditionally administered for the treatment of hepatic diseases, urinary system disorders and diarrhea [9,10].

In recent decades, the utilization of sumac spice is expanding based on increasing empirical evidence that support its ethnopharmacological use and advanced knowledge of the content of nutritionally and medicinally important metabolites such as proteins, unsaturated fatty acids, fiber, and minerals, essential oils, phenolic acids, tannins, anthocyanins and organic acids as indicated in Figure 2. Such evidence has been widely reported from in vitro and in vivo studies, and many have reached the stage of clinical trials in humans [11]. Recent studies are revealing that, sumac is a reservoir of dietary polyphenolic compounds, which has stimulated the study of their phenolic composition and antioxidant properties. Most of the biological activities, health benefits and industrial potential of sumac are increasingly attributed to the predominant polyphenol compounds in the plant materials [3,11]. Furthermore, the volatile constituent (essential oil), which the sumac fame, also comprise bioactive compounds that can significantly contribute to biological activity of the plant [7]. Thus, utilization of sumac is upgrading fast not only in its culinary use but also utilization as pharmaceutical/nutraceutical, food fortifier, food colorant, food preservative and animal feed additive [3,12]. Herein, we discuss the major phytochemical components and antioxidant capacity of *Rhus coriaria* L. with nutritional, medicinal, and industrial significance. To this end, an extensive literature search was performed covering the period up to May 2022 using the MEDLINE/PubMed electronic database (https://pubmed.ncbi.nlm.nih.gov/?term=rhus+coriaria; accessed on 25 June 2022) with the following search strategy: key words “*Rhus coriaria*”. Using the key words, a total of 122 papers were found, among which 93 were published within the last 10 years. Among these 93 were published within the last 10 years. The published materials identified comprise clinical trials (8), Meta Analysis (3), Randomized Controlled Trial (8), and Reviews (9). English-language publications and letters to editors were selected for the study. 

## 2. Sumac as a Functional Food

The statement ‘Let thy food be thy medicine and medicine be thy food’, often ascribed to Hippocrates (400 BC), shows the early recognition of health benefits of healthy diets. The concept of functional food originated in Japan in the early 1980s to refer to processed foods that possess disease-preventing and/or therapeutic properties beyond their basic nutritional roles [11,13,14]. This concept often overlaps with nutraceuticals, which was coined by Stephen DeFelice to emphasize the complementarity between nutrition and pharmacology [13,15]. Sumac fruit has been recognized as a functional food due to the vast proportion of key nutritionally and pharmacologically important constituents such as proteins, essential oils, fatty acids, fiber, minerals, tannins, phenolic acids and anthocyanins [16]. Sumac plant (*Rhus coriaria*) is considered a rich and valuable dietary source of important nutrients which include unsaturated fatty acids, vitamins and minerals. Studies have shown estimates of the overall composition of sumac fruit as illustrated in Table 1 and Table 2.

In a study, Kizil and Turk [12] detected oleic acid (37.7%), linoleic (34.8), palmitic (27.4%), stearic acids (17.3%) and other fatty acids from the petroleum ether extract of the fruit of sumac through GC-MS analysis, indicating the content of unsaturated fatty acids. Analyses of some sumac samples from Iran detected a total of 21 oil compounds representing 86.6% of the fatty acids present in the fruit samples [18]. Proximate analysis of different types of Sumac fruits in Turkey revealed 12.5% oil and 3.5% protein. In this study, Dogan and Akgul [18] further revealed the contents of important fatty acids which include: myristic acid (0.25%), palmitic acid (23.1%), stearic acid (3.1%), and oleic acid (37.5%). Interestingly, the study reported linoleic (omega 6) (34.84%) and α-linolenic acid (omega 3) (1.88%). Alsamri, Athamneh [7] reviewed that, sumac fruit contains important minerals including potassium, calcium, magnesium, phosphorus, iron, sodium, manganese, and copper. The mineral contents were found to be influenced by the prevailing environmental factors and geographic locations [12]. Although, studies on the vitamin composition of *Rhus coriaria* are still fragmentary. A study revealed that, sumac fruit is rich in pyridoxine, ascorbic acid, thiamine and riboflavin [19]. Other vitamins reported include cyanocobalamin, nicotinamide, and biotin [7].

## 3. Bioactive Ingredients in Sumac

The bioactive ingredients in foods, often referred to as the extra-nutritional components, are known to mediate most of their nutraceutical properties as they exert various beneficial pharmacological effects on human health. The fruits of sumac, which are mainly used, are the most well researched part with respect to bioactive ingredients. Other plant parts such as the leaves and seeds have also been reported to hold a number of bioactive compounds. Several phytochemical studies, aimed at determining and identifying the phytocompounds in the *Rhus coriaria*, have revealed over 250 bioactive compounds from various extracts of the fruit, leaf and stem samples of sumac plant. Some of these compounds were detected from aqueous extracts, while others from alcohol extracts and some from lipid extracts. Based on the extensive reports on the bioactive ingredients of sumac, scientific interest for sumac has been growing among researchers and pharmacologists in the recent decade (Figure 2), as it is considered a reservoir of polyphenolic compounds and dietary antioxidants which are known to play critical role in the prevention of various chronic diseases such as cancer, heart disease, and infections [7,11].

Most studies on sumac have revealed similar phytochemical composition, with polyphenols such as tannins, flavonoids and conjugated phenolics as major constituents. The term phenolics refer to phytochemicals comprising aromatic ring with at least one hydroxyl group. Such compounds with one or more aromatic rings possessing more than one hydroxyl groups are referred to as polyphenols (or polyphenolic compounds) as illustrated in Figure 3. These compounds are basically categorized into flavonoids and non-flavonoid compounds. Flavonoids, the dominant class of plant-derived polyphenols comprise two phenolic rings connected by a three-carbon bridge with a common C6-C3-C6 structural skeleton [20]. Non-flavonoids include phenolic acid derivatives, stilbenes, tannins and lignins [21]. In recent decades, tannin-rich plants extracts are becoming popular as additives in ruminant diets to improve the quality of animal products. Plant-derived tannins can be classified into hydrolyzable tannins and condensed tannins. Hydrolyzable tannins contain polyphenol nuclei having molecular weights from 500 to 3000 Daltons (Da), while condensed tannins are oligomeric or polymeric flavonoids comprising flavane-3-ols, including catechin, epicatechin, gallocatechin, and epigallocatechin with molecular weights ranging from 1000 to 20,000 Da [22]. The condensed tannins are not easily degraded by anaerobic enzymes and can depolymerize only with strong oxidation. Several reports which have documented the bioactive components of sumac have revealed hydrolysable tannins as the most prominent group of compounds in the Sumac fruits, followed by flavonoids and phenolic acids [23]. In their study, Abu-Reidah, Ali-Shtayeh [24] detected 211 compounds from methanol extract of sumac fruit samples, with majority belonging to phenolics compounds through an extensive and sophisticated chromatographic analysis. A similar study that employed methanol extract of sumac fruit samples, detected 191 compounds comprising 78 hydrolysed tannins, 59 flavonoids, 9 anthocyanins, and 40 other compounds such as butein [25]. In the case of sumac leaf, analysis of ethereal leaf extract using coloring method identified myricetin alongside gallic acid as the coloring agent of sumac leaves [7,26]. The flavonols quercetin, myricetin, and kaempferol alongside gallic acid, methyl gallate, m-digallic acid, and ellagic acid were identified from ethyl acetate and methanol extracts of sumac leaf samples [7]. Other compounds such as organic acids which include malic, citric, fumaric, and tartaric were also detected from the aqueous extract of sumac fruit samples through HPLC analysis [7]. Most of the compounds have been categorized into hydrolysed tannins, phenolic acids, conjugated phenolic acids, organic acids, anthocyanins, flavonoids, coumarins, xanthones, terpenoids, steroids, and essential oils as depicted in Figure 3. Further, selected bioactive compounds detected in the fruit, leaf and stem samples of sumac are summarized in Table 3. The structures of key polyphenols are depicted in Figure 4.

Efficient extraction methods are needed for the extraction of polyphenols from plant materials. Several reports are available on the isolation and fractionation of phenolic compounds [3,25]. Polyphenols may be extracted from fresh, frozen or dried plant samples. Prior to the extraction process, the plant sample is pre-treated by milling, grinding, drying and homogenization. Application of the drying procedure greatly impacts the total phenolic content. Freeze-drying is known to retain higher levels of phenolic content in the plant samples than air-drying method [31]. A study was conducted in which the extraction of polyphenol from sumac fruit was optimized. The authors varied the conditions in which the extraction took place and reported the optimal extraction conditions as: solvent ethanol (80%); time of extraction (60 min); temperature (40 °C); particle size (1.0 mm); and sumac to solvent ratio 1:15 mL/g [3]. Various analytical methods such as HPLC-MS, UPLC-PDA-ESI/MS, LC-DAD-MS/MS and RRLC-DAD-ESI/MS have been reported to yield several phenolic acids mainly gallic acid and several flavonoids from the ethanolic extracts of sumac fruit samples [25,31,32,33]. 

Essential oils are complex mixtures of low molecular weight compounds, mainly comprising the aromatic and volatile compounds [34,35]. In plants, storage organs of essential oils include oil ducts, resin ducts, glands, or trichomes [36,37]. They can be extracted mainly from the seeds, flowers, peel, stem, bark and whole plants through solvent extraction, steam distillation, and hydrodistillation. Due to their aroma, flavors and natural antimicrobial contents, they are increasingly used in various countries as medicine, perfumes, cosmetics and as food preservatives in food industry [37]. They are increasingly applied as a part of a hurdle technology in which several food preservation factors are combined to provide microbial stability of the food products [38]. *Rhus coriaria* was initially considered as an essential oil-poor plant as the yield of the oil using the available hydro-distillation method was around 0.1% [39]. Several improved analytical techniques which include microwave-aided extraction has greatly increased sumac oil yield to about 13.5% [7,40]. In the study by Gharaei, Khajeh [40], The yield of the essential oil from fruit samples was up to 13.5%, with 21 detected volatile compounds representing 86.6% of the essential oils. Among the volatile compounds β-caryophyllene (30.7%) and cembrene (21.4%) were reported as the main constituents of the essential oil extract. A study showed that, Iranian Sumac essential oils are in its majority β-caryophyllene (30.7%) and cembrene (21.4%) [18]. Kurucu, Koyuncu [39] identified 85 volatile compounds with limonene, nonanal and (*Z*)-2-decenal as the most abundant in essential oil derived from fruit samples using GC-MS. In the same study, 63 volatile compounds were detected showing the most abundant compounds as β-caryophyllene, sesquiterpene hydrocarbons (patchoulane) in the leaf samples, while 63 were identified with β-caryophyllene and cembrene being the most abundant in the stem. Matthaus and Özcan [41] reported linoleic acid, tocopherols and sterols as predominant phytochemicals in essential oil derived from the fruit samples of sumac as detected by HPLC and GLC. A study of fruit samples using gas chromatography techniques revealed the major components of essential oil extracted from sumac fruit samples as β-caryophllene, cembrene, para-anisaldehyde, (*Z*) 2-heptenal, and (*E*)-2-decenal [42]. Analysis of essential oil extracted from sumac fruit, seed, leaf, bud, and flower samples revealed several compounds abundant in the stem as: α-pinene, (*E*)-β-ocimene, limonene, β-pinene, myrcene, (*Z*)-β-ocimene; the leaf samples as: β-caryophyllene and cembrene; and flower samples as: α-pinene and tridecanoic acid [5]. A study aimed at profiling the volatile constituents and organoleptic properties of sumac fruit samples obtained from the southeastern region of Turkey revealed the link between the malic acid constituent of the volatile fraction, accounted and the sour taste of sumac fruit [43]. The authors also revealed other volatiles constituents viz: β-caryophyllene, cembrene, and caryophyllene oxide, which accounted for the flavor of sumac fruits [43]. Volatile composition in plants is known to be affected by various factors such as geographical origin, harvesting time, processing and agricultural practices [44]. A recent study aimed at profiling the volatile compounds of sumac fruit samples obtained from Palestine, Jordan, and Egypt, corresponding to three different geographical locations and origins (Palestine, Jordan, and Egypt) reported 74 volatile components that was grouped into alcohols, aromatics, esters, ethers, furan/aldehyde, hydrocarbons, ketones, monoterpenes, oxides, and sesquiterpene hydrocarbons [4]. Determining the volatile profile of sumac cold tea and post roasting preparation, the authors reported that, in fresh sumac fruit samples sesquiterpenes hydrocarbons were the main volatile compounds class, while furan/aldehydes accounted for the main classes in the roasted fruits. Farag, Fayek [4] further reported that, the volatile profile varied depending on the geographical origin of the samples. For instance, Sumac obtained from Egypt showed remarkable difference among compounds detected from sumac tea sample or roasted sample as compared to those obtained from Palestinian and Jordanian regions, which both revealed similar profiles [4].

## 4. Antioxidant Potential of Sumac

Oxidative stress is an underlying pathophysiological process of several chronic diseases. Reactive oxygen species (ROS) refer to an array of derivatives of molecular oxygen that are normally produced in the system during physiological processes [45]. These reactive species include singlet oxygen, superoxide, hydrogen peroxide (H_2_O_2_), peroxynitrite, hydroxyl and peroxyl radicals. Although, they are ubiquitous and occur as a normal attribute of aerobic life, they are potentially harmful to valuable biomolecules [46]. Uncontrolled and excessive production of ROS causes oxidative stress, which could result in oxidative damage of important cellular components such as proteins and nucleic acids, leading to an impairment of cell function and ultimately drive the pathogenesis and progression of several disease conditions which may include: chronic inflammation, asthma, neurodegenerative diseases, cardiovascular diseases, and cancers [47,48,49]. Dietary polyphenols such as tannins and flavonoids are known to possess strong antioxidant potential by their ability to scavenge free radicals and reactive oxygen species (ROS). These compounds owe their antioxidant activity to their aromatic structural feature, multiple hydroxyl groups, and a highly conjugated system of their chemical structures [20,50]. Such antioxidants are capable of scavenging ROS and delaying or even preventing the irreversible damage of the ROS and resulting oxidative stress to important cell components. Food herbs and spices are known to house large amounts of these antioxidant compounds [20]. 

There are extensive reports on the antioxidant capacity of various extracts and chromatographic fractions derived from the fruits of sumac. For instance, Shafiei, Nobakht [51] revealed the antioxidative, free radical scavenging and lipid peroxidation inhibitory capacity of the methanolic extracts of the fruits of *Rhus coriaria*. These authors also linked the biological activities of the sumac samples with atherosclerosis prevention potential. Another study involving several spice extracts showed that, the aqueous extracts of *Rhus coriaria* exhibited one of the strongest antioxidant capacities as compared to the other spices studied [52]. Using in vitro free radical scavenging activity and antioxidant capacity methods, sumac has shown potent antioxidant potentials. A study showed that, sumac fruit samples significantly inhibited the formation of thiobarbituric acid reactive substances (TBARS) [53]. Another study revealed that, sumac methanolic fruit extract exhibited strong antioxidant scavenging capacity against free superoxide radicals with IC_50_ of 282.92 µg/mL, hydroxyl radicals with IC_50_ of 3.85 g/mL and lipid peroxidation with IC_50_ of 1.2 g/mL in vitro [54]. The aqueous extract of the *Rhus coriaria* fruit samples alone alongside nanoparticles synthesized from the samples exhibited a significant antioxidant activity using ABTS and DPPH assays [55]. Using DPPH and *N*, *N*’-dimethyl-p-phenylendiamina (DMPD) cation radical, sumac aqueous fruit extracts were able to scavenge effectively the radicals showing EC_50_ of 36.4 µg/mL for DPPH and 44.7 µg/mL for DMPD [56]. Sumac leaf extracts showed ABTS radical scavenging activity and ferric-reducing antioxidant power (FRAP) with 725.75 and 41.27 mg Trolox equivalent (TE)/g respectively [57]. 

Studies have attributed the antioxidant capacity of sumac to the constituent polyphenolic compounds. A study revealed that, polyphenol-rich acetone extracts and ethanol extracts derived from fruit samples of sumac showed strong antioxidant capacity [58]. Furthermore, a study revealed that, *Rhus coriaria* had significant antioxidative potential due to its richness in phenolic compounds, especially, gallic acid and its derivatives [59]. Extracts and chromatographic fractions of sumac fruit samples rich in hydrolysable tannins and anthocyanins tannins exhibited strong antioxidant capacity when several extracts and fractions of sumac extracts were screened using ferric thiocyanate and DPPH radical scavenging methods [30]. The authors further showed that, while gallic acid was the most abundant phenolic acid in the fractions, anthocyanin-rich fractions contained cyanidin, peonidin, pelargonidin, petunidin, and delphinidin glucosides and coumarates. A more recent study revealed that, several polyphenols isolated from the *Rhus coriaria* fruit samples exhibited strong in vitro free radical scavenging activities using β-carotene-linoleic acid and 2,2-diphenyl-1-picryl-hydrazyl-hydrate (DPPH) methods as compared to those of glycosides, alkaloids and terpenoids respectively [9]. In a clinical trial, a small amount of GA (in the range of daily consumption in Central Europe) prevented oxidative DNA damage and reduced markers which reflect inflammation and increased risks of cancer and CVD [60]. Gallic acid and other polyphenol are strong antioxidant compounds which are capable of donating hydrogen radicals to the free radicals formed during oxidation becoming radicals themselves that are stabilized by the resonance delocalization of the electron within the aromatic ring. 

## 5. Pharmacological Potential of Sumac

*Rhus coriaria* has been used in homeopathy therapy for thousands of years in the Middle Eastern and South Asian countries [4,61]. It has been traditionally used in the treatment of several illnesses which include liver disease [62], diarrhea [62,63,64], urinary system issues [62], and ulcers [65]. The fruit powder was used to stimulate perspiration [7,64]. The wide range of the medicinal value of sumac may be attributed to the bioactive constituent and antioxidant potentials. Thus, several studies have investigated several extracts, fractions and isolated compounds in disease in order to provide empirical evidence for its ethnopharmacological use. 

### 5.1. Anti-Obesity and Anti-Hyperlipidemic Potential of Sumac

Obesity has been a major risk in the pathogenesis and progression of type-2 diabetes, hypertension, cardiovascular diseases and other cardio-metabolic disorders [66,67,68]. The cardio-metabolic morbidity and mortality due to overweight and obesity observed in the recent decades is increasingly becoming a public health problem of global concern [69,70]. These disorders were only commonly observed in the high-income countries and urban regions characterized with unhealthy diets and sedentary lifestyle. In recent time, scientific reports have revealed an increasing trend in the prevalence of overweight and obesity populations with low socio-economic status [68,69]. Therefore, complementary approaches with anti-obesity potential are sought to prevent or manage obesity. Multiple experimental studies have shown the anti-obesity and antihyperlipidemic effects of sumac extracts and bioactive components. A study reported that, administering sumac fruit extract to rat fed with high cholesterol diet alleviate the lipid abnormalities through significant reduction in serum total cholesterol and triglyceride levels along with augmented activity of serum lactate dehydrogenase and reversed hypertrophic cardiac histology, confirming the anti-obesity and cardioprotective potential of the sumac extracts [51]. A recent study using high-fat diet rats also revealed that, sumac extracts improved hepatic steatosis and dyslipedimia in the experimental animals [71]. The authors further reported that, treatment of the experimental high-fat diet rats with sumac extract led to significant desirable modulatory effects on the serum triglycerides, cholesterol, high-density lipoprotein, low density lipoprotein and malondialdehyde. Experiment involving rabbits showed that, administering sumac extracts led to a significant increase in the platelet distribution width and decreased cholesterol levels in the animals [72]. Methyl gallate, one of the most abundant bioactive compounds in sumac as discussed earlier was revealed as a novel pharmacological stimulator of canonical Wnt/β-catenin signaling, and therefore represents a promising therapeutic agent in obesity [73].

### 5.2. Antidiabetic Potential of Sumac

Diabetes mellitus, a multifactorial metabolic disorder is characterized by sustained hyperglycemia [74,75]. The major pathophysiological events that contribute to diabetes include impaired insulin function, oxidative stress, inflammation, impaired glucose tolerance (insulin resistance) and which eventually results in altered glucose homeostasis and end up in T2DM [76]. Several studies have revealed the antidiabetic potentials of different extracts and fractions derived from sumac samples. These include reports on insulin resistance [77], blood biomarkers [3], activity and gene expression patterns of glucose metabolizing enzymes [3,78]. A report from in vitro analysis revealed that, ethyl acetate fruit extracts from *Rhu coriaria* exhibited antidiabetic potential through inhibition of alpha-amylase [79]. A study showed that, treatment of diabetic experimental animals with hydro-alcoholic extract derived from seed of *Rhu coriaria* led to significant reduction in the glucose and cholesterol levels of the animals [80]. Another study revealed that, diabetic rats treated orally with the lyophilized hydrophilic extract of the sumac fruit samples for a period of 3 weeks led to a significant reduction in the serum levels of glucose, TG, TC, HDL and LDL [81]. The authors further reported that, the sumac extract caused a significant reduction in the level of glycated hemoglobin (HbA1c) and alpha-glucosidase activity while it significantly reduced serum insulin level in the experimental animals. Treatment of diabetic rats with sumac aqueous fruit extracts in another study led to a significant reduction serum glucose, and LDL levels along with reduction in alkaline phosphatase (ALP) activity [82]. Interestingly, several human studies have also revealed the antidiabetic potential of sumac. In a study on the dietary supplementation with *Rhus coriaria* powder in type II diabetic women, significant reduction in serum glucose and insulin resistance index as well as the antioxidant potential of the diet were recorded [83]. The author also reported desirable effects on body weight, body mass index and other anthropometric indices. Another study revealed that, dietary intake of sumac powder (6 g/per day) led to significant reduction in fasting serum insulin level as well as insulin resistance condition in type II diabetic patients [7]. Moreover, a group of patients with type II diabetes daily consuming 3.0 g sumac powder showed a significant reduction in the blood serum glucose as well as the levels of HbA1-c and Apo-B along a significant increase in HbA1-c and total antioxidant levels [84]. Treatment of experimental animals with gallic acid, a prominent compound which belong to hydrolysable tannins group, revealed its in mediating antidiabetic potential by upregulating pAkt, PPAR-γ and Glut4, thus providing potential therapy for diabetes and related disorders [85]. Evidence from cell-based studies show that gallic acid could improve adipose tissue insulin sensitivity, modulate adipogenesis, increase adipose glucose uptake and protect β-cells from impairment [86]. Thus, gallic acid may contribute greatly to the antidiabetic potential of sumac; hence may be valuable the management of obesity-associated type 2 diabetes mellitus. Delphinidin and cyanidin chloride, major anthocyanin constituents of sumac fruits may also account for the antidiabetic potential of sumac they are known to reduce diabetes mellitus complications through reduction of albumin glycation process in vitro and in vivo [87].

### 5.3. Potential of Sumac in Cardiovascular Health

Cardiovascular disease (CVD) causes about a fifth of all deaths worldwide [88]. In the developing countries where 80% of these deaths occur, improving cardiovascular health is important to achieving Sustainable Development Goal (SDG) [89,90]. *Rhus coriaria* is known to contain several phytochemicals that possess antioxidant and antidyslipidemic activity which may help improve cardiovascular health [91]. A study showed that, hydrolysable tannins isolated from the hydro-alcoholic extract of the leaf samples of sumac was able to induce normalization of coronary perfusion pressure, reduction of left ventricular contracture during ischemia, improvement of the maximum rate of rise and fall of left ventricular pressure at reperfusion in male rabbits in a dose-dependent manner [92]. The authors attributed the the cardiovascular protective effect of sumac to COX pathway activation, TNF-inhibition, eNOS activation, and free radical and ROS scavenging. In addition, tannins extracted from the grounded dry leaf sample of *Rhus coriaria* reduced migration of vascular smooth muscle cells (VSMC) by 62% [93]. In patients with dyslipidemia, a study reported that, the endothelial vasodilator function as indicated by the flow-mediated dilation (FMD) significantly improved after consumption of 0.5 g of sumac fruits on daily basis for 1 month [94]. The author further reported that, the systolic and diastolic blood pressure, TC, LDL-C, non-HDL-C, and BMI significantly reduced in the group placed on sumac as compared with the placebo one. In the case of the antidyslipidemic activity of sumac, Sabzghabaee, Kelishadi [95], reported that, sumac *Rhus coriaria* has desirable influence on several lipid profile indices. In this study, significant increase in Apo A-I and HDL was observed alongside with reduction in Apo B, Apo B/Apo A1 ratio, TC, LDL, and TG. Another study revealed that, methanolic extract derived from sumac leaf samples exhibited cardiovascular protective activity in isolated rabbit heart [92]. In this study, the authors reported that, when the rabbit heart was perfused with different concentrations of the methanolic extract, a protective effect against myocardial injury which was induced by ischemia–reperfusion prior to ischemia, was observed in a dose-dependently manner. Beretta, Rossoni [92] further explained that inhibition of creatinine kinase, lactate dehydrogenase as well as activation of endothelial nitric oxide synthase (eNOS) and cyclooxygenases (COX), and scavenging free radicals and ROS may account for the cardioprotective potential of the sumac extract [92]. Furthermore, a study by Anwar, Samaha [96] revealed that, the ethanolic extract derived from sumac fruit samples relaxed isolated rat aorta in a dose-dependent manner. The authors proposed certain mechanisms that may underpin this activity which include PI3-K/Akt signaling pathway, eNOS signaling, nitric and oxide signaling. Other signaling components that may be involved include, soluble guanylate cycle (sGC), cyclic GMP, protein kinase G, cyclooxygesae, adenylyl cyclase/cAMP and ATP-gated potassium channels [96]. These studies provide strong empirical evidence of the cardioprotective potential of sumac. The plant may therefore serve as resources for nutraceuticals and therapeutic agents for ameliorating cardiovascular diseases which include atherosclerosis, aortic aneurysms, and hypertension.

### 5.4. Neuroprotective Potential of Sumac

Neurodegenerative diseases, a cluster of diseases (Alzheimer’s disease, Parkinson’s disease, amyotrophic lateral sclerosis, Huntington’s disease etc.), are characterized by degeneration of the human nervous tissues leading to causing progressive decline in the function of the brain [97]. Parkinson’s disease affects about 10 million people and Alzheimer’s disease affects more than 5.4 million people globally, revealing that these diseases are among the leading causes of death worldwide [97,98]. A recent study investigated the neuroprotective effect of extract obtained from sumac fruit samples through an in vitro cellular model of neuroinflammation [11]. The authors observed that, the sumac extract exhibited a potent anti-inflammatory potential on cell lines through inhibition of ROS nitric oxide; reducing TNFα, iNOS and COX-2 mRNA expression levels; suppressing activation of NFκβ; and enhancement of the transcription of interleukin 10. An animal model, treatment with sumac extracts exerted neuroprotective and anti-inflammatory effects on mouse model of ischemic optic neuropathy [99]. These findings also suggest the potential of sumac in ameliorating neuroinflammation and neurodegenerative diseases.

### 5.5. Role of Sumac in Cancer

Scientific reports have provided strong evidence of the inhibitory role of sumac on tumor growth and survival [100]. El Hasasna, Athamneh [101] revealed the anti-breast cancer activity of sumac extracts using various breast cancer cell lines. The authors reported that, sumac extracts promoted senescence and autophagic cell death, suppressed cell migration, invasion, and metastasis. These observations were later confirmed in vivo using chick embryo tumor growth assay [102]. The authors discussed that, underlying mechanism of the anticancer activity of sumac may involve inhibition of NFκB, STAT3 and NO pathways. They further reported that, sumac showed oncostatic potential in the preventive and therapeutic model of breast carcinoma. Using two vivo models viz: mice 4T1 adenocarcinoma allograft model and chemically induced rat mammary carcinogenesis model, sumac induced strong chemopreventive and therapeutic potential through proapoptotic, antiproliferative, antiangiogenic and epigenetic alterations [103]. In HT-29 and Caco-2 human colorectal cancer cells, treatment with sumac extracts caused significant inhibition of the viability and colony growth of colon cancer cells [104]. In the study, treatment with sumac extract induced Beclin-1-independent autophagy, caspase-7-dependent apoptosis, and inactivation of the AKT/mTOR pathway by promoting the proteasome-dependent degradation of both proteins.

### 5.6. Antimicrobial Potential of Sumac

Several spices are increasingly used to treat infectious diseases or protect food because as they are experimentally proved to possess antimicrobial activities against pathogenic and spoilage microbial agents such as bacteria and fungi [105,106]. Therefore, the extensive reports of the antimicrobial potential of sumac and other spices are not only valuable in the therapeutic field but also desired in the food industry. Increasing resistance of bacteria to many antibiotics constitutes a serious threat in human health [7,107]. Thus, plants rich with ethnomedicinal constitutes and reported antibacterial properties as bioresources for development of novel drugs are indeed receiving more attention. The extensive reports of the antimicrobial potential of sumac and its active constituents are not only valuable in the therapeutic field but also desired in the food industry. In a report by Zhaleh, Sohrabi [108], essential oil derived from *Rhus coriaria* exhibited antibacterial activities against several strains of bacterial. The authors demonstrated the ability of sumac-derived essential oil to reduce the growth of various bacteria which include *Pseudomonas aeruginosa*, *Escherichia coli*, and *Staphylococcus aureus* at concentrations of 2, 3, or 15 mg/mL, respectively. Another study showed that, water extract and ethanol extracts of *Rhus coriaria* showed antibacterial effects against three strains of Pseudomonas aeruginosa and two strains of *Escherichia coli* bacteria on epithelial cells isolated from humans [7]. Methanol extract of sumac exert significant inhibitory actions against the growth of Streptococcus mutant, facultative anaerobic bacterium well known to promote pathogenesis of dental caries and tooth decay [109]. The study further reported that, methyl gallate, a major bioactive constituent of sumac showed a greater inhibitory potential. In another research report, the aqueous extract of sumac inhibited the growth of five common oral bacteria viz: *Streptococcus mutans*, *Streptococcus sanguinis*, *Streptococcus sobrinus*, *Streptococcus salivarius*, and *Enterococcus faecalis* in a concentration-dependent manner [110]. In addition, on orthodontic wire, sumac extract caused significant reduction in the formation of bacterial biofilm by *Streptococcus mutans*, *Streptococcus sobrinus*, *Streptococcus salivarius*, and *Enterococcus faecalis* [110]. Interestingly, the author noted that, the plant extract did not show significant effect against the growth of beneficial bacteria [111]. Thus, *Rhus coriaria* could serve as resource for natural compounds that possess antibiofilm activity, and which could be developed for use in maintaining oral health.

Pathogenic fungi may cause infections in humans, animals or plants. They can infect plant and decrease the growth and yield of crops leading to losses. For instance, *Colletotrichum acutatum* can cause anthracnose in many temperate plants, causing damage to both mature and immature fruits [112]. Research reports have revealed that, sumac’s antifungal potential [113]. A study investigated extracts of *Rhus coriaria* and identified new xanthones viz: 2,3-dihydroxy-7-methyl xanthone, 2,3,6-trihydroxy-7-hydroxymethylene xanthone-1-carboxylic acid, 2-methoxy-4-hydroxy-7-methyl-3-*O*-beta-D-glucopyranosyl xanthone-1,8-dicarboxylic acid, and 2-hydroxy-7-hydroxymethylene xanthone-1,8-dicarboxylic acid 3-*O*-beta-D-glucopyranosyl-(2′-->3″)-3″-*O*-stigmast-5-ene that were all active against fungal infections [114].

## 6. Potential of Sumac in Food Industry

Emerging interests in the food industry include utilization of food additives from natural sources with the aim of developing new products which do not only cater for consumers’ expectations regarding nutritional value and techno-functionality, but also meet the growing need for ‘clean label’ as well as value addition with respect to antioxidant potentials, disease prevention, and health promotion in humans [115]. Thus, the food industry has been undergoing innovative changes and development over the years. This has motivated the use of new food processing methods and natural additives. Utilization of herbs and spices are increasingly gaining prominence not only as flavor enhancers but also as natural preservatives to extend the shelf-life of the products [37,116]. The improvement of nutraceutical properties of foods is also an expected beneficial effect of the inclusion of aromatic plant derivatives to foods and is closely associated with their health-promoting properties [117]. There is an increasing interest in using sumac extracts by the food industry as natural preservatives. A study reported that, water extracts of sumac fruit exhibited a strong antioxidant and antibacterial activity against food-born pathogenic bacteria, suggesting the potential water extracts of the plant as effective and natural preservatives in food manufacturing [52]. Industrially, the seeds are by-products in the production of the spice; however, they are rich in linoleic and oleic acids that qualify the plant seeds to be considered as a valuable raw material for the oil industry. In this context, the mixing of *R. coriaria* seeds oil with olive oil for the use in salads and cooking has already been proposed [24]. Utilization of natural additives, which could meet the growing need for ‘clean label’ and value addition with respect to disease prevention and health promotion in humans, is a current interest in the food industry.

### 6.1. Natural Antioxidant

Oxidation reaction is a main cause of deterioration and spoilage that occurs during food manufacturing and storage. This occurs by several molecular mechanisms such as generation of ROS and free radicals. Oxidation affects many interactions among food constituents, causing both desirable and undesirable products. The lipid components of food are most susceptible to oxidation and resulting in rancidity, which is associated with off-flavors, off-color, altered nutrient profile, and generation of toxic compounds which can hamper the health of consumers. Thus, food producers often utilize antioxidants and chelating agents that, prevent of delay the oxidative process that drive food deterioration. Such antioxidants include butylated hydroxyanisole (BHA) and butylated hydroxytoluene (BHT), which are synthetic with associated toxic and carcinogenic effects [56]. Therefore, natural and safer alternatives antioxidants are sought for in food industry. The antioxidant power of most spices is characterized by their ability to prevent free radical formation, remove radicals, repair oxidative damage, eliminate damaged molecules [118,119]. The most effective antioxidants act via interrupting the free radical chain reactions [120]. A study reported that, the effect of antioxidant capacity of sumac is 50-fold more than vitamins C and E [91]. A study showed that, sumac ethyl acetate extract exhibited higher antioxidant effect at 1% concentration than BHA and BHT and the activity of sumac n-hexane was similar to that of the synthetic compounds [30]. Addition of sumac extract to peanut oil led to reduction in the formation of hydroperoxide and increase oxidative stability of the product in a concentration-dependent manner [121]. The antioxidant capacity of sumac extract has been exploited for improving the quality of sucuk (Turkish dry-fermented sausage) [122]. Fortification of yoghurt samples with aqueous extracts of sumac significantly increased the antioxidant activity of the spiced yoghurt samples in comparison with plain yogurt [57].

### 6.2. Natural Food Colorant

The global natural food colorant market was estimated at USD1144.0 million in 2014, and USD 1625.79 million in 2020, registering a CAGR of 8.47% during the forecast period (2021–2026). Major applications of natural food colorants include confectionary & bakery, beverages, packaged foods, dairy products, frozen foods, condiments, dressings, functional foods and pet foods [122]. The polyphenolic compounds anthocyanins, a main constituent of sumac as discussed earlier are known to contribute mainly to the pigmentation capacity of sumac [3]. These compounds are well reported and utilized pigments which vary in hue from orange, red, blue or purple in color [122]. The stability of these pigments could be enhanced by intermolecular pigmentation after the addition of other polyphenolics, which could interact with the molecule without forming a covalent bond. Therefore, these groups would prevent the nucleophilic attack by water molecule, which makes it a more stable colorant even if mixed with water [122]. Sumac was considered to be similar to wine in its colorant ability. The same compound responsible for the red-like pigmentation was found in both wine and sumac and was identified as hydroxyphenyl pyranoanthocyanins [123].

### 6.3. Natural Food Preservative

The anti-microbial and anti-fungal potential of sumac previously discussed is not only valuable in developing therapeutics but also important in the food manufacturing. Utilization of sumac as a natural preservative may increase the shelf life of the food products and help to maintain its quality [124]. Iranian sumac has been reported to exhibit antimicrobial activity against some food-borne bacteria [125]. Gulmez et al. [124] evaluated the effect of sumac extracts, distilled water and lactic acid on broiler meat with the main aim of improving its microbiological quality and increasing its shelf life. The authors reported that, the effectiveness of sumac extract was comparable to that of lactic acid and higher than that of distilled water making it a good alternative to decontaminate foods other than using synthetic and chemical antimicrobials. Interestingly, sumac-treated wings showed a good color in the sensory evaluation in contrast to both distilled water-treated wings and lactic acid-treated wings which developed an unpleasant color. This gives a positive score for the use of sumac in poultry processing. Treatment of *Botrytis cinerea*, a fungal pathogen that causes grapes table rot, revealed that, sumac leaf extracts were much more efficient than sumac fruits extracts and citric acid treatments [31]. Using different extracts derived from Syrian as a meat tenderizer on *Pectoralis Superficialis* cuts revealed a significant reduction in shear stress and protein content with a significant increase in collagen solubility [126]. The authors also noted a significant decrease in fat content of the product promoting the effect of Sumac as a natural meat tenderizer

### 6.4. Food Fortifier

The polyphenol and antioxidant constituents of sumac could be exploited not only in inhibiting oxidative deterioration of food during processing but also to improve the health benefits of the food products. The use of powder derived from leaf sample of sumac as a fortifier in goat milk yogurt showed a significant increase in total phenolic compounds in comparison with plain yogurt [57]. The authors reported that, the comparison with total phenolic content observed in the sumac water extracts showed that the amount of total phenolic content quantified using the Folin–Ciocalteau method in sumac-fortified yogurt was 53.94%, whereas the remainder remains bound to milk proteins. In this study, regardless of the studied genotype, fortification of yogurt with sumac increased antioxidant capacity than the plain yogurt as indicated by all the assays conducted. Perna, Simonetti [57] further explained the interaction of the protein and polyphenol compounds which may enhance antioxidant capacity. In this way the proteins could entrap more phenolic compounds and therefore increase the stability of polyphenols during the process. Therefore, the potential of using goat milk for the production of fortified fermented products can allow development of new functional foods.

### 6.5. Natural Feed Additive

Natural feed additives are increasingly sought after as the use of synthetic antibiotics as feed additives are banned. Several medicinal herbs are being evaluated as cheaper, safer, healthier alternatives [127]. A study was conducted with the main aim of improving the performance of laying hens and broiler. The authors evaluated the effect of sumac as feed additives in poultry by adding 0.5 and 1% sumac powder to the feed as a supplement and assessed the resultant effect on heat stress in broiler chickens. In the study, the seeds at 0.5% was able to decrease the negative effects of mild heat stress on broiler chickens and increase their efficiency during the first 21 days of age [127]. In a similar study, sumac powder was supplemented in the feed of broiler chickens with the aim of decreasing the thiobarbituric acid reactive substances (TBARS) in the thigh meat of heat-stressed chickens. It was observed that, heat stress caused a decrease in the efficiency of chicken and even cause poultry deaths. Lipid peroxidation as indicated by TBARS levels increased in heat-stressed chicken due to the increase in oxidative stress and lipid oxidation in the skeletal muscles of the chicken. This affects the flavor, the texture, color and nutritional value of the meat, deteriorating its quality. An effort to overcome this was to use different levels of Sumac as a feed supplement for the chicken and compared to Alpha tocopherol acetate as a control. At the end of the experiment, the birds treated with a medium level of sumac showed the lowest thigh TBARS as compared to the non-treated birds [128]. The outstanding pharmacological activity of sumac (*Rhus coriaria* L.) has positioned it as a miraculous spice that can be employed in maintaining health and well-being of both human and animals [129]. In a review by Tong, He [22] several reports were supported the role of plant tannins in the improvement of the quality of animal products such as meat and milk, as well as enhancement of the oxidative stability of the products.

## 7. Safety Considerations

From the search of available literature, sumac has maintained a good track record for safety, with little or no reported adverse effects. However, the fact that sumac belong to the cashew family Anacardiaceae, people with allergies to those foods may want to take caution in the use of sumac to avoid any potential allergic reactions. The poison sumac also known as Toxicodendron vernix belongs to the Anacardiaceae. It produces white-colored fruits, as against the red-hued fruits produced by the edible sumac plant. Poison sumac contains urushiol, the same compound found in poison ivy and poison oak, to which many people. Aguilar-Ortigoza et al. [130] reported that, toxic phenolic compounds occur in 52 species belonging to twenty-seven genera in Anacardiaceae. The authors reported that, majority of these species contain toxic catechols, while a few species contain toxic resorcinols and sixteen species contain biflavonoids. 

## 8. Conclusions

*Rhus coriaria* is a versatile food spice with a wide spectrum of possible uses in human nutrition, pharmaceutical industries and food industries. As discussed in this review, this plant has been widely studied with the aim of expanding its utilization. Various studies have shown the functionality and therapeutic effects of extracts and fractions from this plant which could be attributed to the constituent bioactive compounds. Such plant extracts could be exploited in the search for bioactive natural products that may be useful in the developing new drugs. Polyphenol-rich sumac extracts and its constituent polyphenolic bioactive ingredients could be further exploited towards developing new food products which do not only address consumers’ interests regarding organoleptic, nutritional value and techno-functionality of food, but also meet the growing need for ‘clean label’ as well as value addition with respect to antioxidant potentials, disease prevention, and health promotion in humans. Utilization of sumac in the food industry is increasingly gaining prominence not only as flavor enhancers but also as natural preservatives to extend the shelf-life of the products. More possibilities of sumac extracts could still be explored and exploited on as a preservative and tenderizer for several products like meat and fish. Moreover, polyphenol-rich sumac extracts hold a great potential as natural antioxidant in the production of high-fat foods which are prone to oxidative deterioration during processing. Therefore, polyphenol-rich sumac extracts and the bioactive ingredients could be exploited towards developing new food products which do not only address the growing consumers’ interests regarding organoleptic, nutritional value and techno-functionality of food, but also meet the growing need for ‘clean label’ as well as value-addition with respect to antioxidant potentials, disease prevention, and health promotion in humans. Continued phytochemical characterization and biological evaluation of *Rhus coriaria* in greater details would provide better understanding of its full potential and promote the utilization of this versatile food spice. 

## Figures and Tables

**Figure 1 molecules-27-05179-f001:**
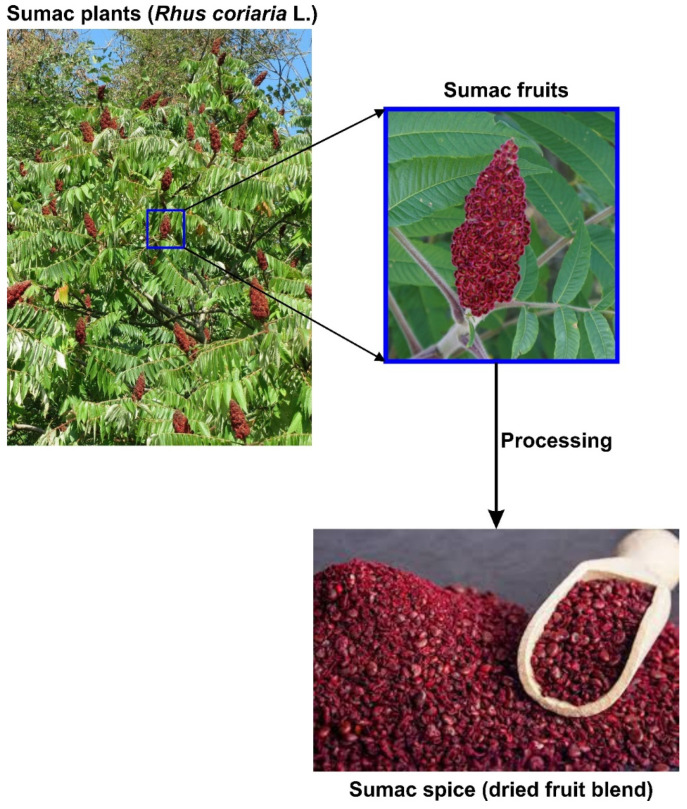
Sumac plants and its spice product.

**Figure 2 molecules-27-05179-f002:**
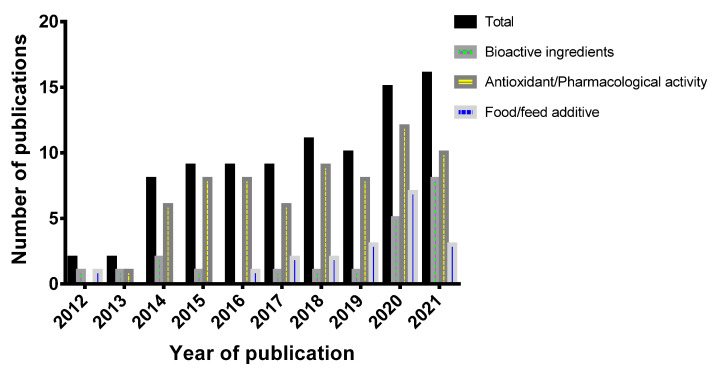
Growing number of publications on the nutraceutical potential of sumac from 2012 to 2021 (https://pubmed.ncbi.nlm.nih.gov/?term=rhus+coriaria; accessed on 25 June 2022).

**Figure 3 molecules-27-05179-f003:**
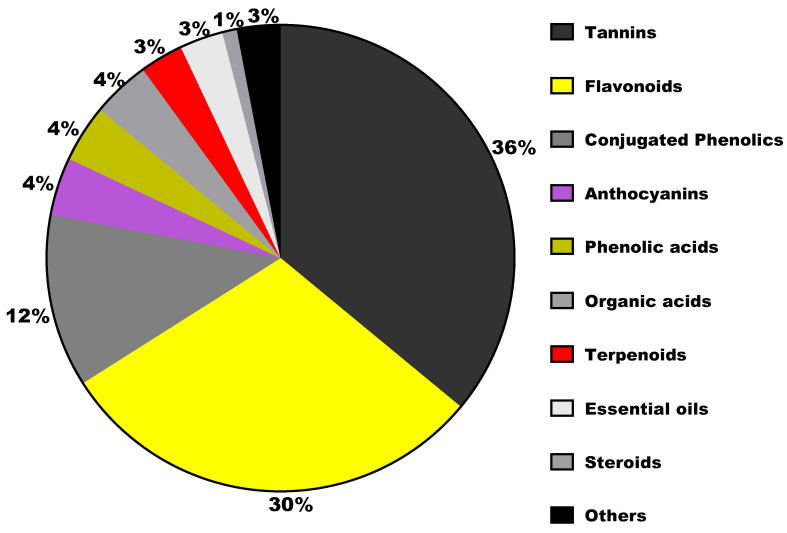
Distribution of 200 phytochemicals from *Rhus coriaria* into sub-classes.

**Figure 4 molecules-27-05179-f004:**
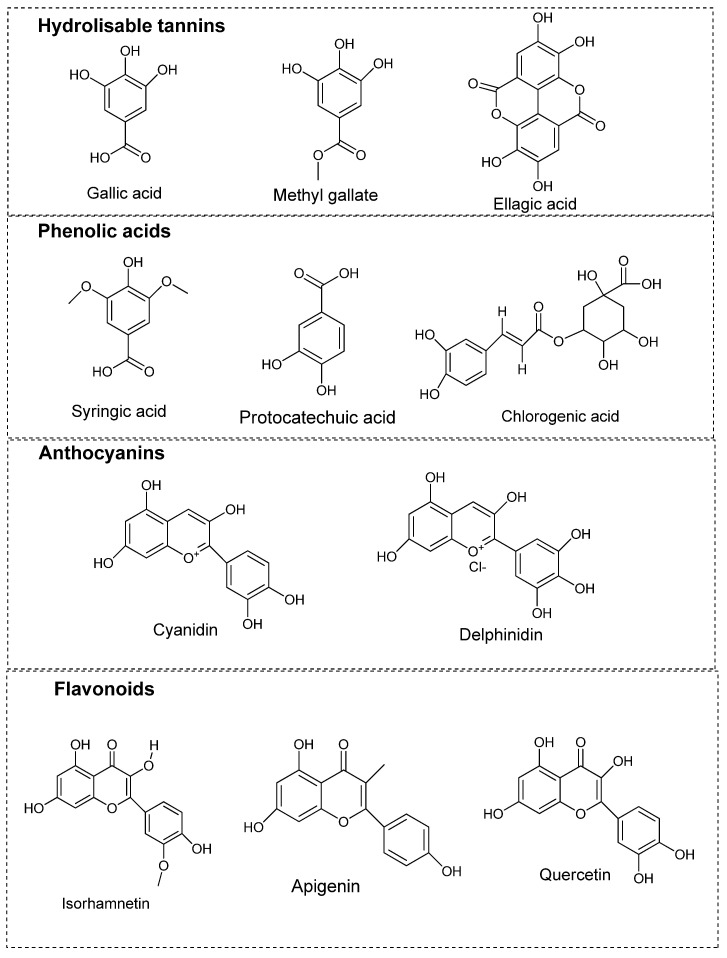
Structures of selected polyphenolic compounds in *Rhus coriaria* L.

**Table 1 molecules-27-05179-t001:** Nutritional composition of the fresh and dried samples of sumac fruits.

Nutritional Components	Fresh	Dried
Moisture (%)	10.60	2.43
Oil (%)	7.40	18.74
Protein (%)	2.60	4.69
Crude fibre (%)	14.60	NDM
Carbohydrate (%)	NDM	71.21
Crude energy (kcal/100 g)	147.8	NDM
Ash (%)	1.80	2.93
Water soluble extract	63.80	NDM
Acidity (%)	4.60	NDM
pH	3.70	3.02

Özcan and Hacıseferoǧulları [3] and Sakhr and El Khatib [17].

**Table 2 molecules-27-05179-t002:** Key nutrients in sumac fruits.

Nutrient	Quanttity
Oleic Acid (%)	37.70
Linoleic Acid (%)	27.40
Palmitic Acid (%)	21.10
Stearic Acid (%)	4.70
Other Fatty acids (%)	9.10
Vitamin B6 (ppm)	69.83
Vitamin C (ppm)	38.91
Vitamin B1(ppm)	30.65
Vitamin B2 (ppm)	24.68
Nicotinamide (ppm)	17.95
Potassium (ppm)	7963.35
Calcium (ppm)	3661.57
Phosphorus (ppm)	1238.74
Magnessium (ppm)	855.95
Iron (ppm)	144.53

Özcan and Hacıseferoǧulları [3].

**Table 3 molecules-27-05179-t003:** Summary of key phytochemicals from *Rhus coriaria*.

Classes of Compounds	Key Bioactive Compounds	Plant Parts	References
Hydrolysable tannins	Gallic acid, methyl gallate, digallic acid, tri-gallic acid, ellagic acid, galloylhexose, *O*-galloylmorbergenin, *O*-galloyl arbutin	Fruits, leaves, seeds	[24,27]
Phenolic acids	Protocatechuic acid, syringic acid, coumaryl-hexoside, caffeoylquinic acid, p-benzoic acid, vanilic acid	Fruits	[9,24]
Conjugated phenolic acid	Galloyl-hexose-malic acid, digalloyl-hexose malic acid, keampferol-hexose malic acid, quercetin-hexose malic acid, Isorhamnetin hexose malic acid	Fruits	[24,28]
Flavonoids	Quercetin, isoquercitrin, quercitrin, rutin, keampferol, myricetin, apigenin, isorhamnetin, isovitexin, rhamnetin, ampelopsin, glycitein-O-glucoside, oxoglycyrrhetinic acid, amenthoflavone, agathisflavone, hinokiflavone and sumaflavone	Leaves, fruits, seeds	[9,24,29]
Anthocyanins	Cyanidin, peonidin, pelargonidin, petunidin, coumarates, delphinidin, myrtillin, crysanthemin	Leaves, fruits, seeds	[24]
Organic acids	Malic acid, citric acid, tartaric acid, linoleic acid, linoleic acid, oleic acid, linolenic acid, palmitic acid, stearic acid	Fruits, seeds	[24,30]
Coumarins	Umbelliferon	Fruits	[24]
Xanthones	2,3-dihydroxy-7-methylxanthone, 2,3,6-trihydroxy7-hydroxymethylene, xanthone-1-carboxylic acid, 2-methoxy-4-hydroxy-7-methyl-3-*O*-beta-D-glucopyranosyl xanthone-1,8-dicarboxylic acid	Leaves, fruits	[9,24]
Terpenoids	Betunolic acid, alpha-tocopherol, tocopherol mannoside, farnesylacetate, pentadecanal, hexadecanal, deacetylforskolin, oxoglycyrrhetinic acid	Leaves, fruits	[4,24,25]
Steroids	Beta-sitosterol	Fruits, seeds	[24]
Essential oils	(*E*)-caryophyllene, n-nonanal, cembrene, alpha-nonoic acid, (2*E*)-decenal, p-anisaldhehyde, (*Z*)-caryophyllene oxide	fruits	[4,9]

## Data Availability

Not applicable.

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
