# Peer review of "Rhus coriaria L. (Sumac), a Versatile and Resourceful Food Spice with Cornucopia of Polyphenols"

_molecules, 2022, doi:10.3390/molecules27165179_

Round 1
Reviewer 1 Report
This paper reviewed the major phytochemical components and antioxidant capacity of Rhus coriaria L. which may account for the nutritional, medicinal, and industrial significance of this important spice. This review is important and interesting and could be accepted with minor revisions.
1. The abstract should be rewritten, too much background has been elaborated. The abstract should focused on summary the content you have reviewed
2. The page number of the references has been missing, Please checking.
3. The Future prospects of Rhus coriaria L. (Sumac) should be elaborated further
4. The extracting method for bioactive compounds should be elaborated further if possible
5. The quality control methods for Rhus coriaria L. (Sumac) and its products should be added if possible
Author Response
Dear Reviewer,
We appreciate your review and comments on our manuscript which greatly improved the quality. Below are responses to the comments:
General Comment: This paper reviewed the major phytochemical components and antioxidant capacity of Rhus coriaria L. which may account for the nutritional, medicinal, and industrial significance of this important spice. This review is important and interesting and could be accepted with minor revisions.
Response: Thank you for the assessment
Comment 1: The abstract should be rewritten, too much background has been elaborated. The abstract should be focused on summary the content you have reviewed
Response: The abstract has been revised accordingly
Comment 2: The page number of the references has been missing, please checking.
Response: The missing page numbers have been added to the references
Comment 3: The Future prospects of Rhus coriaria L. (Sumac) should be elaborated further
Response: The future prospects of the Sumac has been elaborated on in the conclusion section
Comment 4: The extracting method for bioactive compounds should be elaborated further if possible
Response: The extraction of polyphenols in sumac has been elaborated on in the manuscript.
Comment 5: The quality control methods for Rhus coriaria L. (Sumac) and its products should be added if possible.
Response: Additional section that addresses the safety considerations of sumac has been added to the manuscript.
We do hope this will meet up with your criteria for publication.
Thank you.
Reviewer 2 Report
In the presented manuscript, the authors describe the data of the literature on the possibility of multi-directional use of the species Rhus coriaria L. (Sumac), rich in polyphenolic compounds. Especially the authors describe the nutraceutical qualities. The study is interesting. However, the work lacks details of the methodology for developing this literature review - what databases did the authors use? what years? It is very important. In addition, it is worth supplementing the literature with important items, e.g.
Mohit M, Nouri M, Samadi M, Nouri Y, Heidarzadeh-Esfahani N, Venkatakrishnan K, Jalili C (2021). The effect of sumac (Rhus coriaria L.) supplementation on glycemic indices: A systematic review and meta-analysis of controlled clinical trials. Complementary Therapies in Medicine 61: 102766. https://doi.org/10.1016/b978-0-12-819815-5.00048-3
M. Hesam Shahrajabian, Wenli Sun (2022). Using sumac (Rhus coriaria L.), as a miraculous spice with outstanding pharmacological activities. Notulae Scientia Biologicae 14 (1): 1-14. https://doi.org/10.15835/nsb14111118 ...
Author Response
Dear Reviewer
Thank you for your valuable comments and suggestions. Below is our response to the comments raised.
General comments: In the presented manuscript, the authors describe the data of the literature on the possibility of multi-directional use of the species Rhus coriaria L. (Sumac), rich in polyphenolic compounds. Especially the authors describe the nutraceutical qualities. The study is interesting. However, the work lacks details of the methodology for developing this literature review - what databases did the authors use? what years? It is very important. In addition, it is worth supplementing the literature with important items, e.g.
Mohit M, Nouri M, Samadi M, Nouri Y, Heidarzadeh-Esfahani N, Venkatakrishnan K, Jalili C (2021). The effect of sumac (Rhus coriaria L.) supplementation on glycemic indices: A systematic review and meta-analysis of controlled clinical trials. Complementary Therapies in Medicine 61: 102766. https://doi.org/10.1016/b978-0-12-819815-5.00048-3
- Hesam Shahrajabian, Wenli Sun (2022). Using sumac (Rhus coriariaL.), as a miraculous spice with outstanding pharmacological activities. Notulae Scientia Biologicae 14 (1): 1-14. https://doi.org/10.15835/nsb14111118 ...
Response: the details of the methodology and databases employed in developing the review have been added to the manuscript. Also more literature has been included and cited accordingly.
Reviewer 3 Report
The authors presented the review of some functional, physicochemical, and nutritional properties and the Pharmacological potential of sumac as a common food spice.
The idea of this research is relatively good. But unfortunately, it has not been well addressed and some related references have not been reviewed.
Here are some of them:
- Bozkurt H. Investigation of the effect of sumac extract and BHT addition on the quality of sucuk (Turkish dry-fermented sausage). J Sci Food Agricul. 2006;86(5):849–856.
- Fazeli MR, Amin Gh, Ahmadian-Attari MM, et al. Antimicrobial activities of Iranian sumac and avishan-e shirazi (Zataria multiflora) against some food-borne bacteria. Food Contr. 2007;18(6):646–649.
- Fazeli MR, Ashtiani H, Ahmadian-Attari MM, et al. Antimicrobial effect of Rhus coriaria L. (Sumac) total extract on skin isolates Staphylococcus epidermidis and Corynebacterium xerosis. J Med Plants. 2006;5:27–31.
- Kossah R, Nsabimana C, Zhao J, et al. Comparative study on the chemical composition of Syrian sumac (Rhus coriaria L.) and Chinese sumac (Rhus typhina L.) fruits. Pakistan Journal of Nutrition. 2009;8(10):1570–1574.
- Mavlyanov SM, Islambekov Sh Yu, Karimdzhanov AK, et al. Anthocyans and organic acids of the fruits of some species of sumac. Chem Nat Comp. 1997;33:209.
- Mazaheri, T. M., Hesarinejad, M. A., Razavi, S. M. A., Mohammadian, R., & Poorkian, S. (2017). Comparing physicochemical properties and antioxidant potential of sumac from Iran and Turkey. MOJ Food Processing and Technology 5 (2): 288-294.
- Özcan M, Haciseferogullari H. A condiment [sumac (Rhus coriaria L.) fruits]: some physicochemical properties. Bulgarian Journal of Plant Physiology. 2004;30(3-4):74–84.
- Ozcan M. Antioxidant activities of rosemary, sage, and sumac extracts and their combinations on stability of natural peanut oil. J Med Food. 2003;6(3):267–270.
- Rayne S, Mazza G. Biological activities of extracts from sumac (Rhus spp.): a review. Plant Foods Hum Nutr. 2007;62(4):165–175.
- Zalacain A, Prodanov M, Carmona M, et al. Optimisation of extraction and identification of gallotannins from sumac leaves. Biosyst Eng. 2003;84(2):211–216.
Due to the fact that many review articles on sumac have been published so far, so to publish a new review article, we must take a new look at the subject so that the differences between the review research can be clearly seen and can be cited by many researchers in the future.
Unfortunately, I do not see this in this article and therefore I will have to reject this article.
Of course, in my opinion, there is a chance that with the fundamental corrections in this article, you will have the chance to publish it in the future.
I also need to say one thing, and that is that when a review article uses the figures or tables of other articles, permission must be obtained from the relevant publication and referenced (Figures 1 and 3).
To make the work more interesting, I suggest that the authors provide analysis and explain these contradictions, given the numerous articles and sometimes contradictory results reported.
Author Response
Dear Reviewer,
Thank you for the review and the valued comments, which have improved our manuscript. We hope that the responses below will now meet your criteria for publication.
General comments: The authors presented the review of some functional, physicochemical, and nutritional properties and the Pharmacological potential of sumac as a common food spice. The idea of this research is relatively good. But unfortunately, it has not been well addressed and some related references have not been reviewed.
Here are some of them:
- Bozkurt H. Investigation of the effect of sumac extract and BHT addition on the quality of sucuk (Turkish dry-fermented sausage). J Sci Food Agricul. 2006;86(5):849–856.
- Fazeli MR, Amin Gh, Ahmadian-Attari MM, et al. Antimicrobial activities of Iranian sumac and avishan-e shirazi (Zataria multiflora) against some food-borne bacteria. Food Contr. 2007;18(6):646–649.
- Fazeli MR, Ashtiani H, Ahmadian-Attari MM, et al. Antimicrobial effect of Rhus coriaria L. (Sumac) total extract on skin isolates Staphylococcus epidermidis and Corynebacterium xerosis. J Med Plants. 2006;5:27–31.
- Kossah R, Nsabimana C, Zhao J, et al. Comparative study on the chemical composition of Syrian sumac (Rhus coriaria L.) and Chinese sumac (Rhus typhina L.) fruits. Pakistan Journal of Nutrition. 2009;8(10):1570–1574.
- Mavlyanov SM, Islambekov Sh Yu, Karimdzhanov AK, et al. Anthocyans and organic acids of the fruits of some species of sumac. Chem Nat Comp. 1997;33:209.
- Mazaheri, T. M., Hesarinejad, M. A., Razavi, S. M. A., Mohammadian, R., & Poorkian, S. (2017). Comparing physicochemical properties and antioxidant potential of sumac from Iran and Turkey. MOJ Food Processing and Technology 5 (2): 288-294.
- Özcan M, Haciseferogullari H. A condiment [sumac (Rhus coriaria L.) fruits]: some physicochemical properties. Bulgarian Journal of Plant Physiology. 2004;30(3-4):74–84.
- Ozcan M. Antioxidant activities of rosemary, sage, and sumac extracts and their combinations on stability of natural peanut oil. J Med Food. 2003;6(3):267–270.
- Rayne S, Mazza G. Biological activities of extracts from sumac (Rhus spp.): a review. Plant Foods Hum Nutr. 2007;62(4):165–175.
- Zalacain A, Prodanov M, Carmona M, et al. Optimisation of extraction and identification of gallotannins from sumac leaves. Biosyst Eng. 2003;84(2):211–216.
Response: Thank you for the comments. To better address the topic of the review, more literature has been consulted and cited.
Comments: Due to the fact that many review articles on sumac have been published so far, so to publish a new review article, we must take a new look at the subject so that the differences between the review research can be clearly seen and can be cited by many researchers in the future. Unfortunately, I do not see this in this article and therefore I will have to reject this article. Of course, in my opinion, there is a chance that with the fundamental corrections in this article, you will have the chance to publish it in the future.
Response: Thank you for this valuable comment. Although many related articles have appeared online, many of these have not drawn the attention of the audience to the link between the well reported phytochemical profile of sumac and its current utilization in traditional medicine, nutrition, food industry and veterinary practices. Our review aims to fill this gap by pointing out the main phytochemical constituents (polyphenols) as reported in the literature and the multiplicity of its biological activities. We believe that such mechanistic view would improve our understanding of its versatility and greatly promote its future utilization potential.
I also need to say one thing, and that is that when a review article uses the figures or tables of other articles, permission must be obtained from the relevant publication and referenced (Figures 1 and 3).
Response: Copyright permission statement has been requested from the relevant authors.
To make the work more interesting, I suggest that the authors provide analysis and explain these contradictions, given the numerous articles and sometimes contradictory results reported.
Responses: The manuscript has been revised and an additional section, which addresses the safety consideration, is included in the manuscript.
Round 2
Reviewer 2 Report
The manuscript is suitable for publication in its current form.
Reviewer 3 Report
As stated in the previous evaluation, there is still no acceptable innovation in this review article. Besides, it is necessary to compare at least different geotypes. Some articles have not yet been reviewed in this review article, which is considered a weakness for a review article.